

# Comprehensive simulations of new particle formation events in Beijing with a cluster dynamics-multicomponent sectional model

Chenxi Li[1], Yuyang Li[2], Xiaoxiao Li[2], Runlong Cai[3], Yaxin Fan[1], Xiaohui Qiao[2], Rujing Yin[2], Chao Yan[4,5], Yishuo Guo[5], Yongchun Liu[5], Jun Zheng[6], Veli-Matti Kerminen[3], Markku Kulmala[3,5], Huayun Xiao[1]*, Jingkun Jiang[2]*

[1]School of Environmental Science and Engineering, Shanghai Jiao Tong University, 200240, Shanghai, China
[2]State Key Joint Laboratory of Environment Simulation and Pollution Control, School of Environment, Tsinghua University, 100084 Beijing, China
[3]Institute for Atmospheric and Earth System Research / Physics, Faculty of Science, University of Helsinki, 00014 Helsinki, Finland
[4]Joint International Research Laboratory of Atmospheric and Earth System Sciences, School of Atmospheric Sciences, Nanjing University, Nanjing 210023
[5]Aerosol and Haze Laboratory, Beijing Advanced Innovation Center for Soft Matter Science and Engineering, Beijing University of Chemical Technology, 100029 Beijing, China
[6]Collaborative Innovation Center of Atmospheric Environment and Equipment Technology, Nanjing University of Information Science & Technology, Nanjing, 210044, China

*Correspondence to*: Huayun Xiao (xiaohuayun@sjtu.edu.cn), Jingkun Jiang (jiangjk@tsinghua.edu.cn)



**Abstract**

New particle formation (NPF) and growth is a major source of atmospheric fine particles. In polluted urban environments, NPF events are frequently observed with characteristics distinct from those in clean environments. Here we simulate NPF events in urban Beijing with a discrete-sectional model that couples cluster dynamics and multicomponent particle growth. In the model, new particles are formed by sulfuric acid-dimethylamine nucleation, while particle growth is driven by particle

coagulation and the condensation of sulfuric acid, its clusters, and oxygenated organic molecules (OOMs). A variable simulation domain in the particle size space is applied to isolate newly formed particles from preexisting ones, which allows us to focus on new particle formation and growth rather than the evolution of particles of non-NPF origin. The simulation yields a rich set of information including the time dependent NPF rates, the cluster concentrations, the particle size distributions, and the time- and size-specific particle chemical compositions. These can be compared with the field observations to

comprehensively assess the simulation-observation agreement. Sensitivity analysis with the model further quantifies how metrics of NPF events (e.g., particle survival probability) respond to model input variations and serves as a diagnostic tool to pinpoint the key parameter that leads to simulation-observation discrepancies. Seven typical NPF events in urban Beijing were analyzed. We found that with the observed gaseous precursor concentrations and coagulation sink as model inputs, the simulations roughly captured the evolution of the observed particle size distributions; however, the simulated particle growth

rate was insufficient to yield the observed particle number concentrations, survival probability, and mode diameter. With the aid of sensitivity analysis, we identified underdetected OOMs as a likely cause for the discrepancy, and the agreement between the simulation and the observation was improved after we modulated particle growth rates in the simulation by adjusting the abundance of OOMs.



## 1. Introduction


New particle formation (NPF) is frequently observed around the globe and affects the formation of cloud condensation nuclei (CCN) and air quality (Gordon et al., 2017; Kerminen et al., 2018; Lee et al., 2019; Kulmala et al., 2021). NPF events are initiated by the formation of stable molecular clusters by gaseous precursors, followed by the growth of these clusters through condensation and coagulation. The mechanisms of new particle formation and growth (NPFG) are complex. For

instance, particle formation has numerous potential participants that interact with one another (Li and Signorell, 2020; Elm et al., 2020), while particle growth could involve poorly characterized condensables and heterogeneous reactions (Wang et al., 2020; Li et al., 2022; Wang et al., 2010; Kulmala et al., 2022). Additionally, the unfolding of NPF events is critically influenced by the ambient conditions, including the temperature (Frege et al., 2018; Yu et al., 2017), the distribution of pre-existing particles (Deng et al., 2021; Kulmala et al., 2021), and air mass transport (Cai et al., 2018). The complexity of NPFG plus its

sensitivity to ambient conditions has made it a challenge to interpret NPF field observations.

To facilitate the extraction of the underlying mechanisms from observations, particle formation and subsequent growth are often analyzed in an isolated manner. In this isolation, the NPF rate is defined as the particle flux past a size threshold (in practice the instrument detection limit is often used), and the particle growth rate is retrieved by tracking the variation of a representative particle size (Kulmala et al., 2012; Li and McMurry, 2018). The particle formation mechanism is

then obtained by statistically matching NPF models with the observed rates in many NPF events (Jen et al., 2014; Cai and Jiang, 2017; Cai et al., 2021b), and the particle growth mechanism is retrieved in a similar fashion by contrasting the calculated and the observed particle growth rates (Mohr et al., 2019; Qiao et al., 2021). This type of isolated analysis has been instrumental in deciphering the NPFG mechanisms. However, mechanisms thus extracted are statistically averaged, which do not necessarily apply to individual NPF events. Additionally, the prediction of atmospherically relevant quantities, e.g., the

contribution of NPF to CCN sizes, requires the synchronization of particle formation and growth. Therefore, the isolated analyses are ideally followed by their coupling to 'reproduce' the development of NPF events with simulations, which is a stringent test on the applicability of the extracted mechanisms.

The simulation of the evolving particle size distribution (PSD) during NPF events is a built-in feature of some global or regional air quality models. Simulations conducted with these models usually apply large size grids that coarsely simulate

the particle size distributions, focusing on the evaluation of the climatic implications of NPFG rather than the elucidation or verification of their mechanisms (Matsui et al., 2011; Roldin et al., 2019). Some works have applied more elaborate zero-dimensional models to simulate NPF events with greater detail. Huang et al. (2016) combined the WRF-Chem regional chemical transport model and the MALTE-BOX sectional model (Boy et al., 2006) and simulated three representative NPF events at the SORPES station in Nanjing, China. They applied a simplified kinetic nucleation theory to calculate the NPF rates

and showed that the low-volatility products of biogenic vapor oxidation play an essential role in the early growth of freshly formed particles. In a subsequent work, Qi et al. (2018) used a sulfuric acid–highly oxidized molecules (HOMs) NPF scheme to





describe particle formation and compared HOMs' contributions to particle growth at the SMEAR II and the SORPES station. They reported that it was more difficult to reproduce the PSDs observed at the SORPES station possibly due to unaccounted particle growth mechanisms or underestimated condensing organic vapor concentrations of anthropogenic origin. These
detailed simulations provided valuable mechanistic insights into NPFG.

NPF events in urban Beijing are characterized by high NPF rates and comparatively slow particle growth in a polluted environment (Li et al., 2022; Cai et al., 2021b). Compared with NPF events observed in cleaner environments, the PSDs observed in Beijing tend to be more polydisperse due to the long-lasting formation of new particles, and the high concentration of newly formed particles makes coagulation a potentially important mechanism for particle growth (Cai et al., 2021a).
Previously, we have performed isolated analysis of particle formation and particle growth in Beijing, demonstrating that sulfuric acid-dimethylamine (SA-DMA) nucleation governs new particle formation while the condensation of SA and oxygenated organic molecules (OOMs) contributes significantly to particle growth (Cai et al., 2021b; Deng et al., 2020; Qiao et al., 2021; Li et al., 2022; Yan et al., 2021). To date, however, it has not been shown if simulation based on these mechanisms can describe the development of individual NPF events.

In this work, we simulated several NPF events in Beijing with a discrete-sectional model. New particle formation was modelled with cluster dynamics based on the SA-DMA nucleation mechanism, which considered the varying ambient temperature and produced time-dependent cluster concentrations not obtainable from simpler parameterized nucleation rate expressions (Huang et al., 2016; Qi et al., 2018). Particle growth was modelled considering particle condensational growth as well as coagulation, which produced time-dependent PSDs and time- and size-specific particle compositions. Compared to
growth simulation of monodisperse particles (Hodshire et al., 2016; Qiao et al., 2021), this method was particularly suitable for simulating the highly polydisperse PSDs observed in urban Beijing. With the model we assessed to what extent the simulation based on the assumed NPFG mechanism can retrieve the observed evolution of individual NPF events. We further analyzed the likely causes for the simulation-observation discrepancies; towards this goal sensitivity analysis was applied as a diagnostic tool, based on which attempts were made to bridge the gap between the simulation and the observation.

**2. Methods**

**2.1 The discrete-sectional model**

We apply a zero-dimensional model which couples a cluster dynamics module and a sectional module to simulate new particle formation and growth. Our previous work has shown that despite fluctuations in the measured PSD and relevant atmospheric conditions, NPF and subsequent growth in urban Beijing usually occur on a regional scale (Cai et al., 2018), hence
it is reasonable to apply zero-dimensional simulations to analyze the selected NPF events. Air mass transport and primary particle emissions are not incorporated, although their influence on some NPF events is of interest for future investigations. A schematic of the model is shown in Fig. 1. The model considers the formation, growth, and coagulation of both molecular clusters and particles, as well as their loss to pre-existing particles. Input to the model includes the ambient temperature, the





cluster free energies, the time-resolved concentrations of gaseous precursors, and the particle size distribution outside the
simulation domain (explained below).

**Figure 1.**

The cluster dynamics module simulates new particle formation from sulfuric acid (SA) and dimethylamine (DMA). The DMA concentration is assumed to be the same as the $C_2$-amine concentrations observed in the field (Cai et al., 2021b). Collision rate coefficients between molecules and clusters are calculated with the collision kernel given by Chan and Mozurkewich

(2001) with a Hamaker constant of $6.4 \times 10^{-20}$ J, which accounts for the collision enhancement due to van der Waals interactions. The evaporation rates of the clusters are calculated with the collision rate coefficients and the free energy of cluster formation (McGrath et al., 2012), which are available from the literature (Ortega et al., 2012; Myllys et al., 2019; Li et al., 2020). The free energies of $SA_xDMA_y$ clusters at 298 K are taken from Ortega et al.(2012), but the Gibbs free energy of formation of $SA_1DMA_1$ is set to -14.0 kcal/mol at 298 K, which was obtained from our previous work to fit the NPF rates in

Beijing (Cai et al., 2021b). The Gibbs free energy of formation at different temperatures are estimated with Eq. S15 in Cai et al. (2021b). Clusters containing more than 4 SA molecules are treated as nucleated particles and enter the smallest section.

The sectional module simulates the PSD and size-resolved particle compositions as a function of time. The particles are divided into sections according to their volume, and the width of the sections (in terms of the particle volume) increases geometrically by a factor of $2^{0.1} = 1.0718$. Processes considered in the sectional module include particle condensational growth

by $SA_xDMA_y$ ($1 \leq x \leq 4$, $0 \leq y \leq 4$) and OOMs, particle coagulation and loss to pre-existing particles, with the collision rate coefficients between all colliding entities calculated with the collision kernel given by Chan and Mozurkewich (2001). The heterogenous uptake of $HNO_3$ and organic acids by the growing particles is not included in the simulations since they constitute only a minor fraction of particle composition in Beijing according to our previous measurements (Li et al., 2022). More volatile organics may react in the particle phase to form low volatility products and promote particle growth; this process is also not

simulated since it is poorly understood. To track the composition of the particles, each section is further divided into subsections, with one subsection recording the mass of $SA_xDMA_y$ and the other subsections recording the mass of the organic components with different volatilities. Particles in each section are assumed to be internally mixed, i.e., all particles within the same section have the same composition. As a result, when particles coagulate, the chemical composition of the coalesced particles are averaged out by all particles in the section in which they are located.

In the simulation of particle condensational growth, the model treats $SA_xDMA_y$ as non-evaporative, which means that once these clusters condense onto the particles, they do not transfer back into the gas phase. In contrast, to simulate OOMs condensation, the organic vapors are classified into 9 volatility bins, with $\log_{10}C^*$ ranging from -8 to 0 ($C^*$ is the saturation vapor concentration in units of μg m$^{-3}$). Vapors that have higher volatilities are not included since they are not supposed to condense on the freshly formed nanoparticles (Qiao et al., 2021). All vapors that have lower volatilities are classified into the

$\log_{10}C^* = -8$ volatility bin. Extending the number of volatility bins to 11 with $\log_{10}C^* = -10-0$ does not affect the simulation



results (Fig. S6 in the supplemental information, SI). Evaporation rates of OOMs are calculated with their estimated vapor pressures (Qiao et al., 2021), the ambient temperature and their molar fraction in the particle, with the Kelvin effect included in the calculation. The sum of condensation and evaporation rates of $SA_xDMA_y$ and OOMs determines the net mass growth rate of the particles, which is given by

$$\frac{dm_p}{dt} = \sum_i \frac{dm_{p,i}}{dt} = \sum_i m_i(\alpha\beta_i N_i - E_i), \tag{1}$$

where $m_p$ is the particle mass, $m_{p,i}$ is the mass of species i in the particle, $m_i$ is the mass of the i molecules, $\alpha$ is the accommodation coefficient (assumed to be 1 for all species), $\beta_i$ is the collision constant of species i with the particle, $E_i$ is the evaporation rate of species i from the particle. $E_i$ is calculated by

$$E_i = \beta_i N_{i,sat} \exp\left(\frac{4v_i\sigma}{d_p k_B T}\right) f_i \tag{2}$$

where $N_{i,sat}$ is the saturation concentration of species i, $v_i$ is the molecular volume, $\sigma$ is the surface tension of the particle (assumed to be 0.023 N m⁻¹ for all particles (Tröstl et al., 2016)), $d_p$ is the particle diameter, $k_B$ is the Boltzmann constant, $T$ is the ambient temperature and $f_i$ is the molar fraction of species i in the particle. The exponential term in Eq. (2) represents the Kelvin effect. Particle mass change causes the particles to migrate across sections. Both particle number and mass concentrations are conserved during transfer of particles across sections using the method given by Warren and Seinfeld

(Warren and Seinfeld, 1985; Li and Cai, 2020).

Since this work focuses on new particle formation and growth, we apply a variable simulation domain in the particle size space as a function of time to exclude simulating particles that apparently do not originate from the occurring NPF event. This is done by visually examining the measured PSD and setting an upper simulation boundary that encloses the particles formed during NPF with margins (see Fig. 2 for examples of simulation domains). A variable simulation domain helps reduce the

computational cost of the simulation and allow us to focus on NPFG itself rather than the evolution of pre-existing particles or large primary particles. The condensation, evaporation, and coagulation of the particles within the simulation domain are treated explicitly with the methods outlined above, while the particles outside the simulation domain collectively serve as the coagulation sink (CoagS) for the clusters and the particles inside the domain. The CoagS is calculated with the Fuchs equation (Kulmala et al., 2001; Fuchs, 1964 ).

The differential equations of all simulated variables are solved with the MATLAB ode23tb solver. Simulations are conducted in 5-mins intervals, corresponding to the resolution of the field measurements. At the start of each interval, the ambient temperature, SA, DMA and OOMs concentrations are updated to the observed values and are held constant during this interval. Additionally, the collision and evaporation rate constants are updated if the ambient temperature differs from the previous update by 1 K, which ensures that the influence of ambient temperature variation on the rate constants is timely

reflected in the simulation. Simulation results at the end of one interval are used as the initial condition for the next interval.

## 2.2 Field measurement and selected NPF events



The measurement data used in this work were obtained from the ambient observation between Oct. 1st, 2018 to Dec. 31st, 2018 at the AHL/BUCT station. The station is located at the fifth floor of a teaching building in the west campus of Beijing University of Chemical Technology (39°94′N, 116°30′E), which is a typical urban site with three traffic roads and residential buildings nearby within a few hundred of meters (Liu et al., 2020)[24]. State-of-the-art instruments were deployed to measure the key parameters for new particle formation and growth. Specifically, the concentration of $H_2SO_4$ and OOMs were measured with a chemical ionization high-resolution time-of-flight mass spectrometer (HToF-CIMS, Aerodyne Research Inc. and Tofwerk AG) using nitrate ion and its clusters $((HNO_3)_{0-2}NO_3^-)$ as the reagent ions. To quantify $H_2SO_4$ and OOMs concentrations, $H_2SO_4$ sensitivity and m/z-dependent transmission efficiency of the HToF-CIMS were calibrated with published methods (Kürten et al., 2012; Heinritzi et al., 2016). Saturation vapor pressure of OOMs were estimated using the parameterization method in our previous study (Qiao et al., 2021). The concentrations of amines were measured using a modified HToF-CIMS with $H_3O^+$ and its clusters as the reagent ions (Zheng et al., 2015). The aerosol size distribution ranging from 1 nm to 10 μm were measured using a diethylene glycol scanning mobility particle spectrometer (DEG-SMPS, 1-7.5 nm) (Jiang et al., 2011; Cai et al., 2017) combined with a 3 nm -10 μm particle size spectrometer (Liu et al., 2016). More details of the instrument operation, calibration, and quantification of gaseous species and particles can be found in our previous studies (Cai et al., 2021b; Qiao et al., 2021; Yin et al., 2021; Yan et al., 2021).

We selected seven NPF events for analysis based on the availability of measurement data, i.e., the availability of meteorological conditions, the PSD, and the concentrations of SA, DMA, and OOMs. The average temperature, RH, SA, and DMA concentrations between 8:00-18:00 on the seven event days are summarized in Table 1. Out of the seven events, new particle formation and growth were 'fully developed' during events 1-3. In these events, new particle formation was observed with high sub-3 nm particle number concentrations, and the particles grew smoothly during most of the event without abrupt changes (as shown in Fig. 2a). In contrast, during events 4-7 (Figs. S1a in the SI) NPFG seems to be partially influenced by air mass transport or primary particle emissions (as indicated by the sudden disappearance of grown particles at ~14:30 in event 4 and the sudden appearance of ~5 nm particles at ~9:30 in event 5), or sustained particle growth were not observed (events 6 and 7). Since events 1-3 offer the most comprehensive data for comparison with the simulations, in this work the quantitative discussions are focused on events 1-3, but simulation results for events 4-7 are also presented in the SI for completeness.

**Table 1.**

### 2.3 Simulations conducted and comparison metrics

With the numerical model we conduct three types of simulations for the selected NPF events. The first type of simulation is the *base simulation*, in which the input concentrations of SA, DMA, OOMs and CoagS to the model are the same as observed. The second type of simulation is the *improved simulation*, in which we modulate model inputs to improve simulation-observation agreement. The third type of simulation is what we refer to as *5+ simulation* (see the SI). In the 5+ simulation, we





only simulate particle growth above 5 nm, using the observed PSDs below 5 nm as model inputs; this type of simulation is
conducted for diagnostic purposes.

In the comparison of the simulation and the observation, we focus on the metrics listed in Table 2. Brief descriptions of
these metrics are given in Table 2.

**Table 2.**

### 2.4 Sensitivity analysis

We perform sensitivity tests to understand the model response to input variations as well as to diagnose the cause for
simulation-observation discrepancies. In the sensitivity analysis, one of the model inputs, i.e., the SA, DMA, OOMs
concentrations and the CoagS, is scaled by a factor on top of the base simulations while the rest remain unchanged. The scaling
factors are from 0.5 to 1.5 for SA, DMA and CoagS, while the OOMs concentrations are scaled by factors from 0.2 to 5. The
range 0.5-1.5 is a conservative estimate of the SA and DMA measurement uncertainties (Cai et al., 2021b); while the range
0.2-5 covers the OOMs concentration scaling factors applied in the improved simulations and overlaps with scaling factors
used in previous investigations of OOMs' contribution to particle growth (Yan et al., 2022; Tröstl et al., 2016).

As the model inputs are varied, we select four metrics in Table 2 to quantify the model response. The four metrics are
$\bar{r}_{J_{1.4}}$, $\bar{r}_{d_1-d_2}$, $r_{P_{d_1-d_2}}$, and $\bar{r}_{d_m}$. Physically, these four quantities represent the average simulated NPF rate, the particle number
concentration, the average particle survival probability, and the average mode diameter normalized by the respective observed
values. Because of the normalization, metric values close to unity indicate good agreement between the simulation and the
observation.

### 3. Results and discussions

### 3.1 Base simulations

Figure 2 shows the observed and simulated PSDs from 1.5 nm to 100 nm for events 1-3. For all the three events, the
simulation exhibits an overall resemblance to the observation in terms of the PSD shape as well as the timing of NPF. Through
visual inspection, intense NPF was observed between 8:00-13:00, 8:00-16:00, and 9:00-14:30 during events 1-3, respectively
(Figs. 2a). Despite the high coagulation sink on these event days (Table 1), the simulations predict that NPF occurs between
8:00-13:00, 8:00-17:30, and 9:00-15:30, respectively (Figs. 2b). In event 1, the simulated NPF timing is a perfect match with
the observation; in events 2 and 3, the simulation slightly overestimates the NPF duration. The PSDs for events 4-7 are shown
in Fig. S1. For events 4 and 5, the simulated NPF time window largely overlaps with the observation, although the
discrepancies between simulated and observed PSDs seem larger than those of events 1-3 possibly due to processes not
considered in the model (i.e., air mass transport or primary particle emissions); for events 6 and 7, both the observation and
the simulation show new particle formation without sustained particle growth.

**Figure 2.**





230       We next quantitatively compare the simulation and the observation with respect to the *NPF rates*, the *SA dimer concentration*, the *particle number concentration*, the *particle survival probability*, and the *particle mode diameter*. Figures 3a compare the simulated and observed NPF rates at an electrical mobility diameter of 1.4 nm for events 1-3. Figures 3a show that the simulated rates differ from the observation to various extents. In event 1, the simulated and the observed rates are very close, with the value of $J_{1.4,sim}/J_{1.4,obs}$ between 0.8 and 2; in events 2 and 3, $J_{1.4,sim}/J_{1.4,obs}$ lies within 3-10 and 3-8,

respectively. Considering the high uncertainties of both NPF rate measurement and modelling, the agreement of $J_{1.4,sim}$ and $J_{1.4,obs}$ is fair and their discrepancies are within the ranges reported by previous works which compared simulated and observed NPF rates (Cai et al., 2021b; Jen et al., 2014; Kürten et al., 2018). Figures 3b compare the simulated and the observed SA dimer concentrations. The ratio $N_{d,sim}/N_{d,obs}$ lies in the range of 4-5, 0.9-1.1 and 2-4 for events 1-3, respectively, which are also within ranges reported by our previous work on NPF rates in urban Beijing (Cai et al., 2021b).

240       Figures 3c-3d compare the simulated and observed particle number concentrations in the size range $1.5 - 3$ nm ($N_{1.5-3}$) and between 5 nm and the reference curve ($N_{>5}$), respectively. The simulated $N_{1.5-3}$ is similar to or higher than the observation (Figs. 3c), but the simulated $N_{>5}$ is lower than the observation (Figs. 3d). This flip of order means that a higher percentage of particles are lost during growth in the simulation than in the observation. Figures 3e quantitatively compare the simulated and observed particle survival probability from 1.5 nm to 3 nm, from 3 nm to 5 nm, and from 5 nm to 8 nm. The simulated particle

survival probability is lower than the observation for all the three events, but the underestimation is less an order of magnitude. A few previous works have discussed new particle survival in Beijing (Kulmala et al., 2017; Tuovinen et al., 2022; Cai et al., 2022b). Kulmala et al. (2017) showed that for sub 3 nm particles the theoretical survival probability could be several orders of magnitude lower than the observation in polluted megacities; more recently, Tuovinen et al. (2022) showed that for Beijing this discrepancy can be smaller but is still 1-2 orders of magnitude at high CS/GR (GR stands for growth rate) conditions. In

this context, the disagreement of the simulation and the observation is not large although there is room for improvement. Since particle survival probability is mainly governed by GR/CoagS, the lower simulated particle survival probabilities suggest underestimated GR, overestimated CoagS or both in the simulation (Kerminen and Kulmala, 2002; Cai et al., 2022b).

**Figure 3.**

      In terms of the mode diameter $d_m$, Figs. 3f compare the simulated and the observed $d_m$ between 14:30 and 16:30, during

which the mode diameter is clearly identifiable. The $d_m$ after 16:30 is not considered because in some events the PSDs go through abrupt changes (as shown clearly at 17:00 in Fig. 2, a3) with increased particle number concentrations in the mode, possibly attributed to air mass transport, primary particle emissions, or the shrink of the atmospheric boundary layer. Figures 3e show that the simulated particle mode diameters are a few nanometers smaller than the observation (the averaged difference is 4.9, 6.7 and 4.2 nm for events 1-3, respectively). The smaller simulated $d_m$ could indicate low simulated GR but could also

be influenced by other factors such as the delayed end of simulated NPF which may shift $d_m$ to smaller sizes at a given time (particles formed later in an event has shorter growth time and contribute smaller particles to the mode).



The analysis above suggests that the discrepancy between the simulation and the observation could have convoluted origins. Other than the inaccuracy of the assumed NPFG mechanism (which is beyond the scope of this study), a possible cause for the discrepancy is that the model inputs, i.e., the concentrations of SA, DMA, OOMs and the CoagS, deviate from their actual values since they are subject to non-negligible measurement uncertainties (Cai et al., 2021b; Qiao et al., 2021). To further pinpoint the cause for the discrepancy, we next tune the simulation inputs systematically to understand the model response to input variations.

### 3.2 Sensitivity analysis

Figure 4 shows the response of $\bar{r}_{J_{1.4}}$, $\bar{r}_{>5}$, $r_{P_{3-5}}$ and $\bar{r}_{d_m}$ (see Table 2) to variations of model inputs for event 1. We remind the readers that $\bar{r}_{J_{1.4}}$, $\bar{r}_{>5}$, $r_{P_{3-5}}$ and $\bar{r}_{d_m}$ are the average NPF rates, the average >5 nm particle number concentrations, the particle survival probability from 3 nm to 5 nm, and the average mode diameter normalized by the observed values. Similar plots for events 2 and 3 are shown in Fig. S2 in the SI.

**Figure 4.**

Figure 4a indicates that $\bar{r}_{J_{1.4}}$ and $\bar{r}_{>5}$ increase substantially as SA concentration increases, with variations of SA by $\pm 50\%$ leading to changes of $\bar{r}_{J_{1.4}}$ and $\bar{r}_{>5}$ by 1-2 orders of magnitude. The high sensitivity of $\bar{r}_{J_{1.4}}$ and $\bar{r}_{>5}$ with respect to the SA concentration is expected since the NPF rates is a strong function of SA concentrations in polluted regions with high CoagS(Cai et al., 2021b). In contrast to $\bar{r}_{J_{1.4}}$ and $\bar{r}_{>5}$, the particle survival probability $r_{P_{3-5}}$ is a weak function of SA when it is scaled between 0.5-1.5. The weak dependence occurs for two reasons. First, compared to OOMs, SA is a minor contributor to particle growth in event 1 (shown later in section 3.4), hence its variation within a modest range does not strongly influence GR/Coag. Second, higher SA concentrations lead to higher particle number concentrations as well as particle number consumptions by coagulation. Consequently, as the SA concentration increases, the coagulational loss of particle numbers partially offsets the effect of higher GR on survival probability. Figure 4a lastly shows that $\bar{r}_{d_m}$ is more sensitive to SA at higher SA concentration, which means the mode diameter is more affected by SA variations when SA is a more important contributor to particle growth.

Figure 4b shows that, compared with SA, increasing DMA concentration has a modest effect on $\bar{r}_{J_{1.4}}$ and $\bar{r}_{>5}$. The weaker dependency is explained by the overall weaker dependence of the NPF rates on DMA concentration during NPF events in Beijing (Cai et al., 2021b). The DMA concentration barely influences $r_{P_{3-5}}$ and $\bar{r}_{d_m}$ because DMA does not directly participate in particle growth.

Figure 4c shows that as the OOMs concentration increases, $\bar{r}_{J_{1.4}}$ decreases slightly but $\bar{r}_{>5}$, $r_{P_{3-5}}$ and $\bar{r}_{d_m}$ increase considerably. Increasing OOMs concentration can significantly promote particle growth in event 1 as OOMs are the main contributors to particle growth in this event (shown later in section 3.4). Consequently, $\bar{r}_{J_{1.4}}$ decreases as OOMs increase the surface areas of particles which scavenge $SA_xDMA_y$ clusters and suppresses NPF. Simultaneously, higher OOMs leads to higher GR and GR/CoagS which increases $\bar{r}_{d_m}$, $\bar{r}_{>5}$ and $r_{P_{3-5}}$. Interestingly, the scaling of OOMs concentration has a





converging effect on the sensitivity curves towards unity in the vicinity of 1.5 (shaded in purple) in Fig. 4c, corresponding to a good agreement of the four selected metrics. A similar converging effect on $\bar{r}_{d_m}$, $\bar{r}_{>5}$ and $r_{P_{3-5}}$ by OOMs scaling is also seen in events 2 and 3 with scaling factors of approximately 4 and 2, respectively (Fig. S2 in the SI). Such convergence of comparison metrics implies that adjusting OOMs concentrations is probably more effective to bring the simulation closer to the observation than adjusting other model input parameters for which similar converging effects are not seen. The difference between the curve-converging scaling factors might be partially caused by the different ambient conditions on the event days; for events 1-3, the scaling factor is larger for colder days (Table 1).

Figure 4d shows the effect of CoagS on the simulation. The increase of CoagS strongly decreases $\bar{r}_{J_{1.4}}$, $\bar{r}_{>5}$ and $r_{P_{3-5}}$ because the CoagS frustrates particle formation and survival by scavenging clusters and particles. Although CoagS does not directly affect the particle growth rate, $\bar{r}_{d_m}$ becomes large at low CoagS values. This occurs due to enhanced coagulational growth as low CoagS leads to high particle number concentrations and promotes particle coagulation. Compared to SA, DMA and OOMs, the CoagS has a lower measurement uncertainty of 10% (Cai et al., 2021b); however, in the calculation of the CoagS, including collision enhancement effect by long range interactions (e.g., van der Waals forces) can increase the CoagS by 30-40% percent (Chan and Mozurkewich, 2001; Cai et al., 2022a). In this work we have used the Fuchs equation to calculate CoagS (Kulmala et al., 2001), which is in line with most of the previous works but does not consider the effect of collision enhancement. Figure 4d suggests that scaling CoagS by a factor of 1.3-1.4 significantly decreases particle number concentrations and worsens the agreement between the simulation and the observation. With the limited number of NPF events examined in this work, it is difficult to conclude on the appropriate functional form of the CoagS for simulating NPF events.

We note that at a CoagS scaling factor of 0.75, the selected metrics also converge to ~1 in Fig. 4d, suggesting adjusting CoagS could narrow the gap between the simulation and the observation for event 1 (similar to OOMs scaling). However, the analysis above suggests it is more likely that the CoagS is underestimated in our simulation due to the neglect of coagulation rate enhancement. Additionally, similar converging effects are not obvious for the other two events (Fig. S2). Therefore, the convergence shown in Fig. 4d is likely to be fortuitous.

Overall, Fig. 4 demonstrates that the simulation is sensitive to the model input parameters, which implies that if systematic errors exist in measurements, albeit moderate and unavoidable from the perspective of measurement (e.g., an $\pm 50\%$ uncertainty in condensable concentrations), the simulated quantities could change by more than an order of magnitude. Figure 4 and Fig. S2 also reveal that the OOMs is somewhat unique among the inputs because scaling their concentration to higher values leads to the convergence of the sensitivity curves. This hints that it is possible that the organic condensable vapors were underdetected during the field observations.

In a previous work (Qiao et al., 2021) we have shown with single particle growth simulations that the condensation of $SA_xDMA_y$ and OOMs can (on average) explain the particle growth in spring, summer and autumn in Beijing during 2018-2019, but was insufficient to explain the particle growth in winter (events 2 and 3 occurred in winter). The low simulated particle growth in winter was tentatively attributed to that the observed condensable organic vapor concentration was





systematically biased low at low temperatures, or that heterogeneous growth processes (e.g., oligomerization) were neglected (Qiao et al., 2021). Additionally, it has been shown that the nitrate CI-APi-ToF is mainly sensitive to highly oxygenated organic molecules (oxygen number >5), thus it may underestimate low and semi volatility organic compounds due to the lower ionization efficiencies (Riva et al., 2019; Hyttinen et al., 2015); scaling their concentration accounting for the weaker ionization
efficiency can effectively improve the growth simulation (Tröstl et al., 2016). Based on the probably underdetected condensable vapor concentrations suggested by these previous works as well as the converging effect of OOMs scaling in the sensitivity analysis, we next adjust the OOMs concentration in the simulation to modulate particle growth, with the goal to improve the simulation-observation agreement. Note that although heterogeneous processes are not considered in the simulation (mainly because these processes in the newly formed particles are poorly understood), OOMs concentration
amplification may have similar enhancing effects on particle growth as incorporating heterogeneous reactions which leads to the formation of low volatility products.

### 3.3 Simulations with improved parameters

OOMs concentrations were adjusted on top of the base simulations so that the simulated and the observed particle mode diameters agree between 16:00-16:30. We refer to the simulations with OOMs concentration adjustment as the *improved*
*simulations*. Figure 5 compares the improved simulations with the observations for events 1-3, while Fig. S3 shows the comparison for events 4-5. For events 6-7 the OOMs concentration was not adjusted since well-defined mode diameters do not exist in these short-lived NPF events.

**Figure 5.**

Figures 5h show that the simulated and the observed $d_m$ almost overlap in the improved simulations. To achieve this
agreement, the OOMs concentrations were scaled by factors of 1.35, 4 and 1.8 in events 1-3, respectively. Comparison of Figs. 5a and 5b indicates that the improved simulations still roughly capture the timing of NPF and better captures the shape of the PSDs compared to the base simulations. Figures 5c and 5d show that the NPF rates and the SA dimer concentration in the improved simulations are very close to the base simulations (Figs. 3a and 3b), which is expected since OOMs do not directly affect new particle formation in the model. Figures 5e and 5f show that although the simulated $N_{1.5-3}$ are close to the base
simulations (Figs. 3c), the $N_{>5}$ in the improved simulation is significantly higher than the base simulation (Figs. 3d) and is closer to the observed values. In terms of the particle survival probability, Figs. 5g indicate that the gap between simulated and observed particle survival probability is narrowed after OOMs concentration adjustment, with the simulated and the observed $P_{3-5}$ and $P_{5-8}$ almost the same. Apparently, the increase of OOMs concentration in the simulation increases GR, which leads to better agreement of the simulated and observed particle survival probability.
Despite the improved agreement, discrepancies between the simulation and the observation still exist. One notable discrepancy is that the simulated $P_{1.5-3}$ is lower than the observation, in particular for events 2 and 3 (Figs. 5, g2 and g3). The low simulated $P_{1.5-3}$ could be caused by the insufficient simulated GR in the 1.5 – 3 nm range. Particles in this size range are





subject to strong Kelvin effects, hence their growth are mainly subject to the concentration of extremely low volatility vapors. An underestimation of the concentration of such vapors even in the improved simulations may have led to slow particle growth

between 1.5 nm and 3 nm. Alternatively, an overestimation of the Kelvin effect (which leads to overestimated vapor pressure at the particle surface and hence overestimated evaporation rate) or the neglect of the heterogenous reactions that produce low volatility products in the particle phase may also cause low simulated GR. Apart from insufficient GR, another plausible explanation for the low simulated $P_{1.5-3}$ (or the high observed $P_{1.5-3}$) could be that the particle number concentration in the 1.5 nm-3 nm range are underdetected compared to those in the >3 nm range, which causes the observed $P_{1.5-3}$ to be higher than

they really are. Lastly, primary emissions of particles above 3 nm can also elevate the observed $P_{1.5-3}$ above their real values, which is not considered in the simulation.

A second discrepancy between the simulation and the observation is the delayed rise of the simulated $N_{>5}$ compared to the observed $N_{>5}$ for event 2 (by about 1 hour) and event 3 (by about 2 hours), as shown in f2 and f3 of Fig. 5. To identify the underlying cause for this discrepancy, we conducted 5+ simulations on top of the improved simulations (see section S2). In

the 5+ simulations we only simulated particle growth above 5 nm and used the observed sub 5 nm PSDs as model input. As shown in Fig. S4, the $N_{>5}$ from the 5+ simulations closely follows the observation, which suggests that (a) at the conditions of the improved simulations, the model can describe particle growth and loss of the ≥5 nm relatively well, and (b) the delay in Fig. 5f largely originates from the inability of the model to reproduce the observed PSD the in the ≤ 5 nm size range. The reason behind this 'inability' could be complex. First, it is possible that the freshly formed particles grow too slowly in the sub

5 nm size range to allow ~ 5 nm particles to appear in time in the simulations. Second, there might be < 5 nm particles formed elsewhere transported to the observation site which offer a starting point for particle growth, leading to the early appearance of > 5 nm particles in the field observations, but such transport of particles is unaccounted in the simulation. Third, the NPF rates might be underestimated in the simulation at the start of the NPF events but correctly simulated later, which causes an 'uneven' appearance of > 5 nm particles. A revisit to Fig. 5c gives a hint of the validity of this hypothesis. The ratio $J_{1.4,sim}/J_{1.4,obs}$

should be lower at the early stage of the events if this hypothesis is correct. As shown in Figs. 5c by the fitted ratios (i.e., the black solid line), $J_{1.4,sim}/J_{1.4,obs}$ is not ostensibly lower at the start of events 2 and 3; hence this hypothesis is unlikely to be the cause of the delay.

To summarize this section, with moderate OOMs concentration adjustment we have been able to narrow the gap between the simulation and the observation in terms of particle number concentration, particle survival probability and particle mode

diameter simultaneously, which implies that the OOMs concentrations might have been overall underdetected during field observations and contributed to the disagreement in the base simulations. However, we do not rule out the possibility that the neglect of heterogeneous growth process in the base simulation also contributed to the gap shown in section 3.1; if this is the case, the scaling of OOMs concentration could be interpreted as a compensation for this missing growth mechanism. We also note that with a combined tuning of multiple model input parameters (i.e., not limited to OOMs concentration scaling) or

compound-specific OOMs scaling (e.g., based on the O/C ratio of the compound), even better simulation-observation agreement might be achieved. Here we restrain parameter tuning to uniform OOMs concentration adjustment to avoid over-



interpreting the simulation results with only a limited number of NPF cases. More systematic investigations to identify the underlying cause for simulation-observation gaps can be facilitated by analyzing a larger NPF dataset with improved OOMs measurements.

### 3.4 Particle compositions

An advantage of the current simulation compared to single particle growth models is the retrieval of the particle chemical composition for any particle size at any time from the simulations. We next examine (1) particle composition variation as a function of particle size at a fixed time, and (2) particle composition variation as a function of time at fixed particle sizes. The former reveals information on the major participants of particle growth at different particle sizes, while the latter is relevant to
particle composition field measurements with instruments such as the TDCIMS (Smith et al., 2010; Li et al., 2022).

**Figure 6.**

Figures 6a and 6b show the simulated particle mass composition as a function of particle size at 13:00 for the base and the improved simulations. For the three events, $SA_xDMA_y$ dominates the composition of small particles, while OOMs take up higher mass fractions in larger particles, indicating more organics contribute to particle growth as the Kelvin effect decreases
with particle size. At small particle sizes (e.g., sub 3 nm) the particle composition varies strongly as a function of particle size, but this variation gradually levels off as the particle size further increases. The nearly constant particle compositions at larger particle sizes indicate that the Kelvin effect itself does not significantly influence particle compositions for above-3 nm particles. This is in agreement with our previous report that the size-dependence of particle composition for 8-40 nm new particles is not simply caused by the Kelvin effect, but also the variations of precursor concentrations with time (Li et al.,
2022). Additionally, the variation of particle composition with particle sizes is not unique to a specific time; the plots of the particle composition at 11:00 have similar trends and are shown in Fig. S5.

OOMs contribution to particle growth differs for the three events. In the base simulations (Figs. 6a), OOMs mainly drives particle growth in event 1, $SA_xDMA_y$ mainly drives particle growth in event 2, and the contributions of $SA_xDMA_y$ and OOMs to particle growth are comparable in event 3. In the improved simulations (Figs. 6b), OOMs still dominates particle growth in
event 1 but its contribution to particle growth becomes on par with $SA_xDMA_y$ in event 2. The contrast of particle composition between a2 and b2 of Fig. 6 demonstrates the importance of constraining the simulation with particle composition measurements, as it is conceivable that measurement-constrained particle sulfate to organics ratios could support or oppose adjusting OOMs concentration by a factor of 4 in event 2. In fact, the composition in the improved simulation is closer to our recent field measurements of 8-40 nm new particle compositions where organic compositions always dominate (Li et al., 2022).

Figures 6c show the composition variation of 2 nm, 8 nm, and 15 nm particles as a function of time in the improved simulation of event 2. The same plots for events 1 and 3 are shown in Fig. S5. The fraction of $SA_xDMA_y$ in the particle has an overall decreasing trend with time (the same trend is observed for events 1 and 3 as shown in Fig. S5). This occurs because the SA concentration usually reaches its peak earlier than the OOMs concentration in urban Beijing; consequently, $SA_xDMA_y$



takes up a higher mass fraction of the particles that appear early in the day (Li et al., 2021; Li et al., 2022). Figures 6c
additionally show that the variation of $SA_xDMA_y$ fraction in four hours is less than 25% for both 8 nm and 15 nm particles,
and the fractional change within a time span of half an hour is less than 4%. This level of particle composition change indicates
that TDCIMS measurement, which typically collects particles at fixed sizes for no more than half an hour during NPF events
in Beijing (Li et al., 2022), should have a composition measurement uncertainty of no more than a few percent attributable to
particle composition variation with time.

**4. Conclusions**

We simulated the development of several NPF events in urban Beijing with a discrete-sectional model. NPF formation
by SA-DMA nucleation was simulated with a cluster dynamics module, while particle growth was simulated considering vapor
condensation and particle coagulation. A variable simulation domain was applied, which enabled the isolation of new particle
formation and growth from the evolution of larger particles of non-NPF origin. With a set of selected metrics, e.g., the NPF
rates and the particle survival probability, the simulation was comprehensively assessed by comparison with the observation.
We additionally designed sensitivity analysis and targeted simulations (i.e., the 5+ simulations) to trace the cause for
simulation-observation discrepancies.

With the observed gas precursor concentrations as model inputs (i.e., the base simulations), we found that the simulation
can roughly capture the development of several selected NPF events which were not apparently influenced by unaccounted
processes (e.g., air mass transport). The base simulations underestimated the particle growth rates in these events, which led
to lower-than-observed particle number concentrations, survival probabilities and particle mode diameters. Sensitivity analysis
was then conducted to identify the cause for the discrepancy. The analysis suggested that the simulation could be sensitive to
model input uncertainties. For instance, in event 1 an $\pm50\%$ variation of SA and CoagS could cause more than an order of
magnitude differences in simulated particle formation rates and number concentrations, while increasing the OOMs
concentration considerably promoted particle growth. The sensitivity analysis also showed that OOMs scaling had a
converging effect on the sensitivity curves and implied that the OOMs concentration might have been underdetected during
the field observations. With additional rationale supported by our previous work on new particle growth in urban Beijing, we
conducted improved simulations with scaled OOMs concentrations and were able to narrow the gap between the simulation
and the observation. Further analysis of particle chemical composition showed that the organic fraction was significantly
increased in the improved simulations for event 2; such a change can be coupled with field measurements of particle
compositions to constrain the actual condensable concentrations during the NPF events.

While most of the work on NPFG focus on the statistical analysis of many NPF events, this work analyzed NPFG in detail
with an event-based approach. This approach is complementary to the statistical method and demonstrates to what extent
individual event simulated with an assumed NPFG mechanism agrees with the observations. Both approaches have their
strengths and should be conducted in the analysis of NPF field observations if feasible.





**Code and data availability:** The simulation data and the Matlab code that produces all figures in the manuscript are available from the corresponding authors upon request.

**Author Contribution:** CL and JJ initialized the study. CL developed the model and did the simulations. YL, XL, RC, XQ, RY, CY, YG, YL, JZ, VK, MK, JJ supported the study with field measurements and data analysis. CL took the lead in writing the manuscript and the other authors contributed to the writing and revision of the manuscript.

**Competing Interests:** Some authors are members of the editorial board of Atmospheric Chemistry and Physics. The peer-465 review process was guided by an independent editor, and the authors have also no other competing interests to declare.

**Acknowledgments:**

This research has been supported by Natural Science Foundation of Shanghai (grant no. 21ZR1430100), the special fund of State Key Joint Laboratory of Environment Simulation and Pollution, National Natural Science Foundation of China (grant 470 no. 22188102 and 92044301), Samsung PM$_{2.5}$ SRP, and the Academy of Finland (332547).

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





**Table 1.** Case numbers, dates, the concentrations of SA and $C_2$-amine (assumed to be equal to the DMA concentration in this work), the condensation sink of $SA_1DMA_1$, the temperature and the RH for the 7 selected NPF events. Note that the concentrations, CS, temperature, and RH are the average values between 8:00 to 18:00 on each event day.

| Case number | Date YYYY.MM.DD | SA ($\times 10^6$ cm$^{-3}$) | $C_2$-amine (pptv) | CS (0.001 s$^{-1}$) | Temperature (°C) | RH (%) |
|---|---|---|---|---|---|---|
| 1 | 2018.10.27 | 2.48 | 3.97 | 11.7 | 15.0 | 19.1 |
| 2 | 2018.12.13 | 2.68 | 1.34 | 5.79 | 2.7 | 15.1 |
| 3 | 2018.12.18 | 3.26 | 1.56 | 9.81 | 8.3 | 19.1 |
| 4 | 2018.10.28 | 1.91 | 2.93 | 3.19 | 16.2 | 25.6 |
| 5 | 2018.10.30 | 1.68 | 2.52 | 6.69 | 15.4 | 23.4 |
| 6 | 2018.11.07 | 1.43 | 7.47 | 16.5 | 9.4 | 30.2 |
| 7 | 2018.12.26 | 2.22 | 1.28 | 8.74 | -3.3 | 14.1 |



**Table 2**. Metrics for comparison between the simulation and the observation, along with the method to calculate these metrics.

| Metric | Description |
|---|---|
| $J_{d_p}$ | The new particle formation rate at a threshold diameter $d_p$, calculated by applying a particle population balance formula (Eq. S1 in the SI) to the simulated or the observed PSDs. |
| $N_d$ | The SA dimer concentration, calculated by adding the concentrations of all $SA_2DMA_y$ ($0 \leq y \leq 4$) clusters in the simulation or in the field observation. |
| $N_{d_1-d_2}$ | The particle number concentration in the diameter range $[d_1, d_2]$, obtained by integrating the number-based PSDs from $d_1$ to $d_2$. |
| $P_{d_1-d_2}$ | The particle survival probability from $d_1$ to $d_2$, calculated by dividing the time-integrated $J_{d_2}$ by time-integrated $J_{d_1}$ (Eq. S2 in the SI). |
| $d_m$ | The mode diameter, determined by locating the local maxima of the PSD in the $dN/d\log_{10}d_p$ form. |
| $r_{P_{d_1-d_2}}$ | The ratio of the simulated and the observed $P_{d_1-d_2}$. |
| $\bar{r}_{J_{1.4}}$ $\bar{r}_{d_1-d_2}$ $\bar{r}_{d_m}$ | The ratios of the *average* $J_{1.4}$, $N_{d_1-d_2}$, and $d_m$ between the simulation and the observation. |



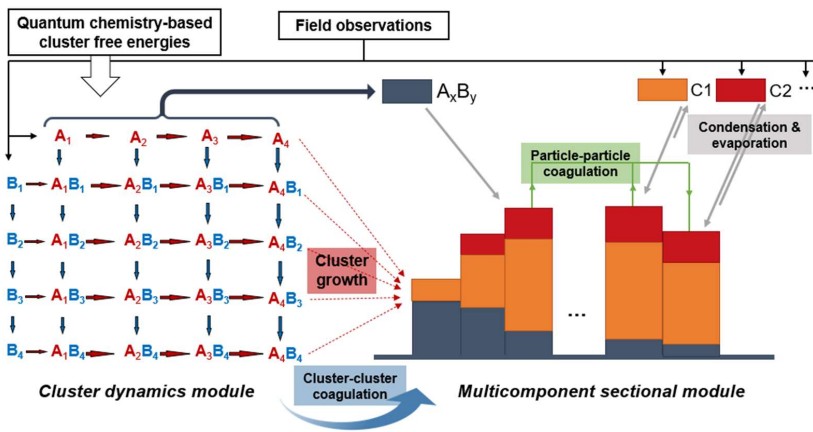

**Figure 1**. A schematic of the simulation model used in this work. A and B represent acid and base molecules, respectively; C1 and C2 represent two condensing organic vapors of different volatilities. Cluster formation by cluster-cluster association is not shown in this figure but is included in the simulations.




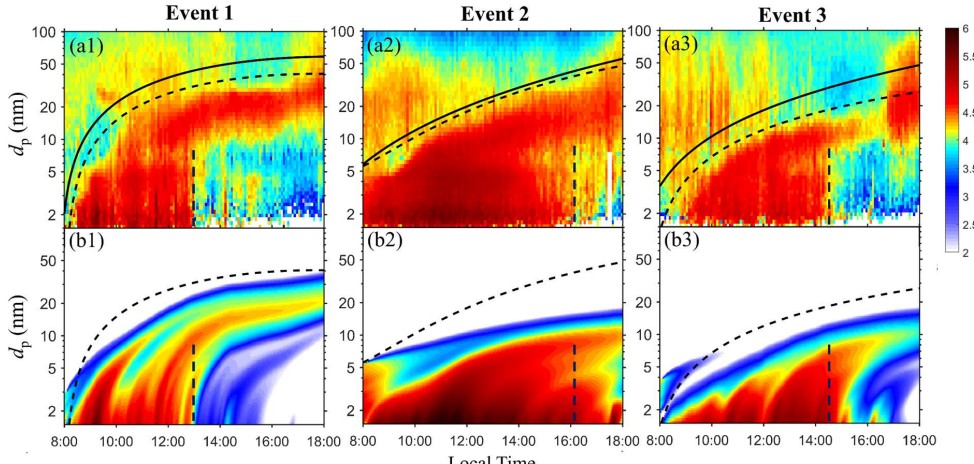

**Figure 2**. Comparison between the observed and the simulated PSDs in events 1-3. The color bar shows the $\log_{10}$ values of the particle size distribution ($dN/d\log_{10}d_p$ in units of #/cm$^3$). (a) The observed particle size distribution. (b) The simulated particle size distribution. The black solid curves in Figs. 2a are the variable simulation domain boundaries applied in the simulation (see section 2.1). Two additional visual guides are also plotted to facilitate the simulation-observation comparison. The vertical dashed lines approximately mark the end of the observed NPF, and the black dashed curves are reference curves which mark the upper boundary of the observed PSDs.

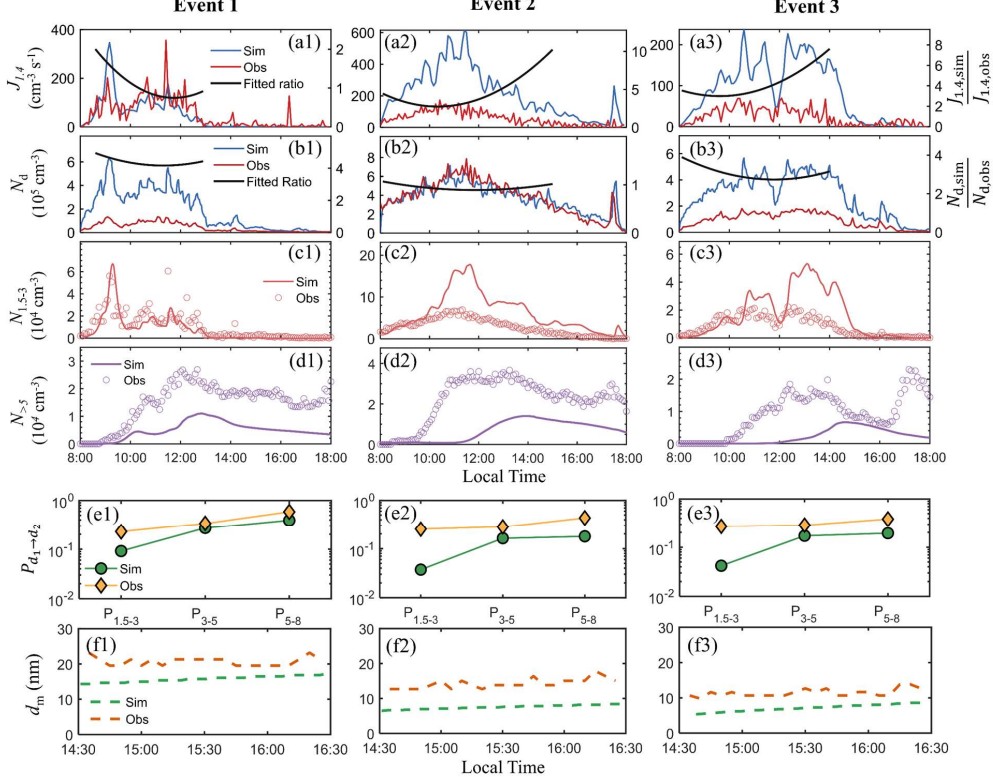

**Figure 3.** (a) The simulated NPF rates $J_{1.4,\mathrm{sim}}$, the observed NPF rates $J_{1.4,\mathrm{obs}}$ (left axis) and their ratios $J_{1.4,\mathrm{sim}}/J_{1.4,\mathrm{obs}}$ fitted to second order polynomials (right axis). (b) The simulated SA dimer concentration $N_{\mathrm{d,sim}}$, the observed SA dimer concentration $N_{\mathrm{d,obs}}$ (left axis) and their ratios $N_{\mathrm{d,sim}}/N_{\mathrm{d,obs}}$ fitted to second order polynomials (right axis). (c), (d) The simulated and observed particle number concentrations between 1.5-3 nm and between 5 nm and the reference curves (see Fig. 2). (e) The simulated and the observed particle survival probability from 1.5 nm to 3 nm ($P_{1.5\text{-}3}$), from 3 nm to 5 nm ($P_{3\text{-}5}$) and from 5 nm to 8 nm ($P_{5\text{-}8}$). (f) The simulated and the observed mode diameters between 14:30 and 16:30.





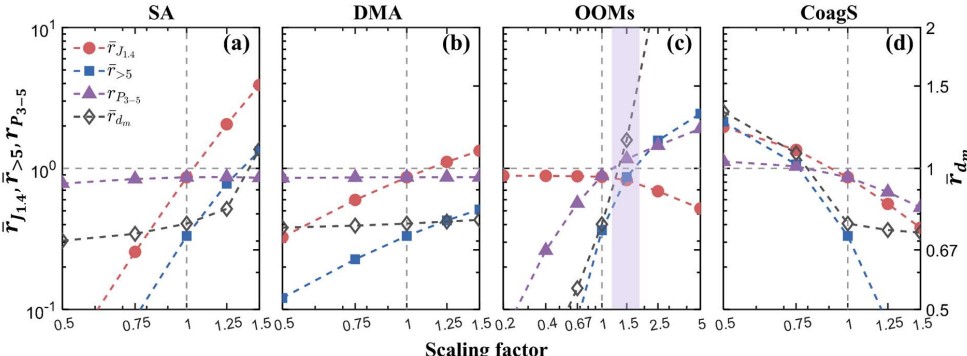

**Figure 4**. The sensitivity of $\bar{r}_{J_{1.4}}$, $\bar{r}_{>5}$, $r_{P_{2-5}}$ and $\bar{r}_{d_m}$ (see Table 2 for explanations of these quantities) to the scaling of SA, DMA and OOMs concentrations and the CoagS for event 1. In the calculation, $\bar{r}_{J_{1.4}}$, $\bar{r}_{>5}$ were calculated with values averaged from 8:00 to 18:00, while $\bar{r}_{d_m}$ was calculated with values averaged between 14:30 and 16:30. $\bar{r}_{J_{1.4}}$, $\bar{r}_{>5}$ and $r_{P_{2-5}}$ correspond the left y-axis, and $\bar{r}_{d_m}$ corresponds to the right y-axis. Two reference lines are shown to aid visualization: the horizontal dashed line corresponds to a ratio of unity, while the vertical dashed line corresponds to the base simulation condition. The purple shade in (c) approximately includes the OOMs scaling factors which leads to the convergence of the sensitivity curves.









**Figure 5**. (a) The observed particle size distribution. (b) The simulated particle size distribution. In (a) and (b), the black dashed curves are reference curves that enclose the upper boundary of the observed PSDs which appear to originate from NPF. Vertical dashed lines approximately mark the end of the observed NPF events. (c) The simulated NPF rates $J_{1.4,\,\mathrm{sim}}$, the observed

715 NPF rates $J_{1.4,\,\mathrm{obs}}$ (left axis) and their ratios $J_{1.4,\mathrm{sim}}/J_{1.4,\mathrm{obs}}$ fitted to second order polynomials (right axis). (d) The simulated SA dimer concentration $N_{\mathrm{d,\,sim}}$, the observed SA dimer concentration $N_{\mathrm{d,\,obs}}$ (left axis) and their ratios $N_{\mathrm{d,sim}}/N_{\mathrm{d,obs}}$ fitted to second order polynomials (right axis). (e), (f) The simulated and observed particle number concentrations between 1.5-3 nm and between 5 nm and the reference curves. (g) The simulated and the observed particle survival probability from 1.5 nm to 3 nm, from 3 nm to 5 nm and from 5 nm to 8 nm. (h) The simulated and the observed mode diameters between 14:30 and 16:30.

720



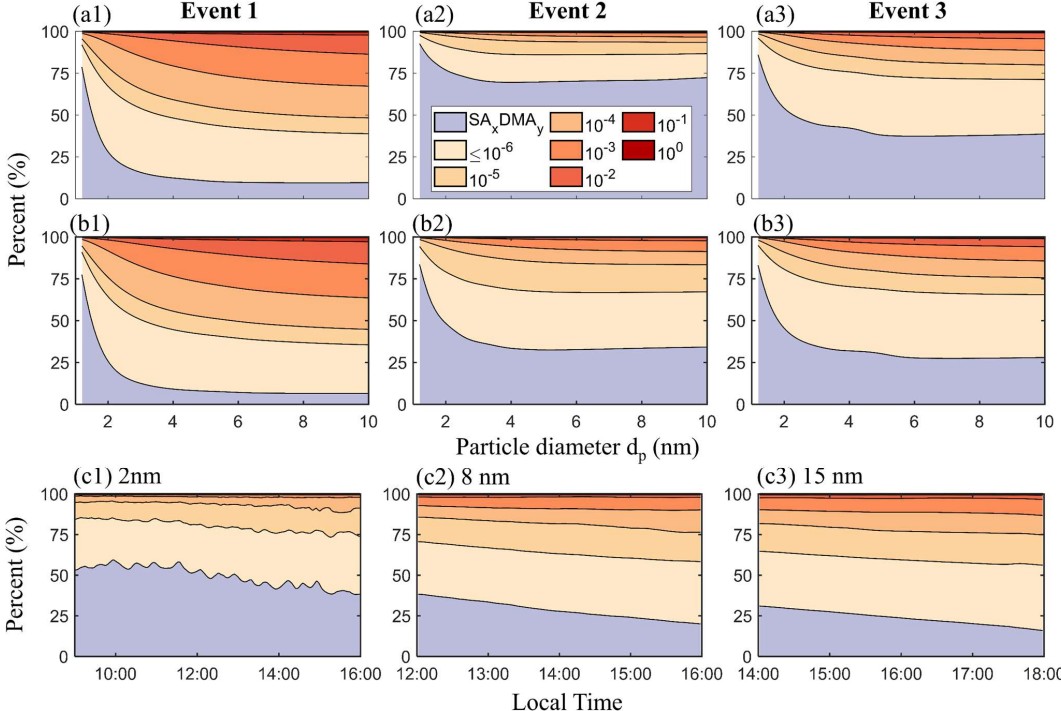

**Figure 6**. (a), (b) Composition of the particles smaller than 10 nm at 13:00 for events 1-3. (a) and (b) correspond to the base simulations and the improved simulations, respectively. (c) Particle composition variation with time in event 2 at fixed particle sizes (2 nm, 8 nm, and 15 nm) in the improved simulations. The organic species with $C^* \leqslant 10^{-6}$ µg/m$^3$ are binned together and labelled '$\leqslant 10^{-6}$' in the figure legend.