# Peer review of "Comprehensive simulations of new particle formation events in Beijing with a cluster dynamics-multicomponent sectional model"

_Atmospheric Chemistry and Physics, 2022_

## Referee Comment (RC1)

This study simulated new particle formation (NPF) and growth events in Beijing with a discrete-sectional model that couples cluster dynamics and multicomponent particle growth. Through comparison with the field observations, the study have done a comprehensively assessment on the simulation-observation agreement. Further sensitivity analysis with the model also quantified how NPF respond to model input variations. The topic of this study fits the scope of Atmospheric Chemistry and Physics, and such study are quite valuable to improve our understanding on new particle formation and growth processes. I recommend it can be accepted after the following revisions.

**Major issues:**

In this study, new particle formation is considered from sulfuric acid (SA) and dimethylamine (DMA), is this only one mechanism caused nucleation events during the observation period? Do you think any other nucleation schemes (e.g. binary, ternary) or any other species such as NH3 may contribute to the observed NPF in Beijing?

**Minor issues:**

Line 48. The authors stated that temperature critically influenced NPF events. But in other measurements, the mean temperature in NPF and non-NPF days was almost identical (Yan et al., 2021). Can you talk about this inconsistency?

Line 151-154: 'Since this work focuses on new particle formation and growth, we apply a variable simulation domain in the particle size space as a function of time to exclude simulating particles that apparently do not originate from the occurring NPF event. This is done by visually examining the measured PSD and setting an upper simulation boundary that encloses the particles formed during NPF with margins (see Fig. 2 for examples of simulation domains)'. This is key technique point, suggest use Fig as example to provide more details or explanations on how to do this variable simulation domain?

Figure 3. can you also plot the simulated Amine concentration and compare with measurements?

Figure 4. I suggest to use 'n' to represent the average >5 nm particle number concentrations, rather than 'r'

Figure 6. please clearly state what are these compositions shown in the figures, the authors only give SAxDMAy and the organic species with C\* $\leq$ 10-6 µg/m3, what are other colors?

Line 258. Should Figures 3e be Figures 3f?

Line 345. Can you explain why the OOMs concentrations were scaled by factor of 1.35, 4 and 1.8 in events 1-3? Are these numbers the typical OOMs concentrations from measurements in polluted areas?

Line 704. Should  $r_{P2-5}$  be  $r_{P3-5}$ ?

---

## Author Comment (AC1)

**Responses to Referees' Comments on Manuscript egusphere-2022-748**

**Comprehensive simulations of new particle formation events in Beijing with a cluster dynamics-multicomponent sectional model**

We are grateful for the referees' comments and these comments has helped us to improve the manuscript. Please find our point-to-point responses below. Comments are shown as *blue italic text* and the revised texts are shown as "quoted underlined text". In the revised manuscript, the changes are highlighted. The line numbers in the response refer to the revised manuscript without tracked changes.

**Referee 1**: This study simulated new particle formation (NPF) and growth events in Beijing with a discrete-sectional model that couples cluster dynamics and multicomponent particle growth. Through comparison with the field observations, the study have done a comprehensively assessment on the simulation-observation agreement. Further sensitivity analysis with the model also quantified how NPF respond to model input variations. The topic of this study fits the scope of Atmospheric Chemistry and Physics, and such study are quite valuable to improve our understanding on new particle formation and growth processes. I recommend it can be accepted after the following revisions.

**Major issues:**

1. In this study, new particle formation is considered from sulfuric acid (SA) and dimethylamine (DMA), is this only one mechanism caused nucleation events during the observation period? Do you think any other nucleation schemes (e.g., binary, ternary) or any other species such as NH3 may contribute to the observed NPF in Beijing?

Response: We thank the reviewer for the valuable comment. It is possible that other nucleation mechanisms contribute to NPF, e.g., SA-NH3, SA-NH3-DMA, SA-Organics. However, in the polluted environment of Beijing, Cai et al. (2021) has shown that the NPF rates in Beijing can be well explained by the SA-DMA nucleation without invoking other mechanisms. The fast NPF by SA-DMA is due to the comparatively high concentrations of amines and the high stability of SA-DMA clusters (compared to e.g., SA-NH3 clusters). Other species in addition to sulfuric acid and amines may also contribute to nucleation in the complex urban atmosphere, yet we found minor or negligible contribution from either SA-NH3 or SA-Organics. As to SA-NH3-DMA, simulation with ACDC using the thermodynamic data from Li et al. (2020) shows that only a small fraction of small acid-base clusters contain NH3 at typical conditions in Beijing (see Fig. S9 in Yin et al. (2021); if we use the cluster thermodynamic data computed by Myllys et al. (2019) instead, the nucleation rate is simply too low to explain the particle formation rates in Beijing). In the current work, it is shown in Fig. 3a that the simulated NPF rates with the SA-DMA mechanism alone is on par with or higher than the observed rates. Therefore, unless new observational evidence suggests otherwise, we think it is proper to only consider SA-DMA nucleation.

We note the choice to use SA-DMA mechanism alone is somewhat unique to polluted urban environments, e.g., Beijing and Shanghai (Yao et al., 2018; Cai et al., 2021). In other

environments where DMA play a lesser role in clustering with SA (Bianchi et al., 2016; Yan et al., 2018), other mechanisms should be included in NPF rate calculations.

**Revised manuscript, lines 108- 110**: "Although other binary or ternary nucleation mechanisms (e.g., SA-NH3, SA-organics, SA-NH3-DMA) might contribute to NPF in some environments, here we only consider SA-DMA because our previous work has shown that the NPF rates in Beijing can be well explained by the SA-DMA nucleation without invoking other mechanisms (Cai et al., 2021)."

**Minor issues:**

2. Line 48. The authors stated that temperature critically influenced NPF events. But in other measurements, the mean temperature in NPF and non-NPF days was almost identical (Yan et al., 2021). Can you talk about this inconsistency?

**Response:** The nucleation rates depend on several factors, including the concentration of the nucleating species, the temperature, and the condensation/coagulation sink. For a given concentration of nucleating species (e.g., SA, DMA) and condensation/coagulation sink, temperature variation can strongly alter the NPF rates by influencing cluster stabilities. However, in the real atmosphere the temperature changes along with other NPF influencing variables. For instance, the OOMs concentration is typically higher in the summer than winter (Qiao et al., 2021). If we consider the SA-organics NPF mechanism, although temperature increase leads to decreased clusters stability, the more abundant nucleation precursors can compensate the influence of temperature.

The results presented by Yan et al. (2021) demonstrate the strong influence of CS on the occurrence of NPF, but do not suggest temperature does not influence NPF from a mechanistic point of view. This is because the influence of temperature on NPF is not isolated from other variables.

3. Line 151-154: 'Since this work focuses on new particle formation and growth, we apply a variable simulation domain in the particle size space as a function of time to exclude simulating particles that apparently do not originate from the occurring NPF event. This is done by visually examining the measured PSD and setting an upper simulation boundary that encloses the particles formed during NPF with margins (see Fig. 2 for examples of simulation domains)'. This is key technique point, suggest use Fig as example to provide more details or explanations on how to do this variable simulation domain?

**Response**: We thank the reviewer for the suggestion. To make the procedure of picking simulation domain clearer, we revised the manuscript as follows,

**Revised manuscript, lines 156 -160**: "This is done by first picking 10-20 time-size points in the pseudo color plots of the measured PSD (i.e., Fig. 2a). These points are above the upper end of particle size distribution originated from NPF by some margins. A second order polynomial is used to fit these points, i.e., to obtain the particle size as a function of time. The fitted polynomial is subsequently used to calculate the simulation boundary at a given time in the simulations (i.e., solid black lines in Fig. 2a)."

4. Figure 3. can you also plot the simulated Amine concentration and compare with measurements?

**Response**: The amine concentration is taken from observation and used as inputs to the simulation. The average amine concentrations during the NPF events are available in Table 1.

5. Figure 4. I suggest to use 'n' to represent the average >5 nm particle number concentrations, rather than 'r'

**Response:** We chose to use 'r' because Figure 4 shows the *ratios* of the simulation to the observation rather than the absolute number concentrations. Therefore, we think it is more appropriate to keep using 'r'.

6. Figure 6. please clearly state what are these compositions shown in the figures, the authors only give  $SA_xDMA_y$  and the organic species with  $C^* \leq 10-6 \mu g/m3$ , what are other colors?

**Response:** The other colors correspond to organics with different volatilities. To make this clear, we added a sentence in the caption of the figure:

**Revised manuscript, lines 747-748**: "The color-species/volatilities relation is shown in (a2), where the numbers correspond to the volatility of the organics in units of  $\mu$ g/m3. Note that the organic species with C\* $\leq$ 10-6  $\mu$ g/m3 are binned together and labelled ' $\leq$ 10-6 ."

7. Line 258. Should Figures 3e be Figures 3f?

Response: Yes, we thank the reviewer for catching this mistake.

8. Line 345. Can you explain why the OOMs concentrations were scaled by factor of 1.35, 4 and 1.8 in events 1-3? Are these numbers the typical OOMs concentrations from measurements in polluted areas?

**Response**: These factors are obtained by fitting the simulated mode diameter to the observation, so these factors are fitting parameters to account for the possible under-detection of OOMs by nitrate-CIMS. After scaling, the average condensable OOMs concentration are  $4.13 \times 10^7$  cm-3,  $1.12 \times 10^7$  cm-3 and  $1.79 \times 10^7$  cm-3 for events 1-3, respectively. By comparing to Fig 2 in Qiao et al. (2021), it is clear that even after scaling, these concentrations are within or close to the typical ranges of OOMs observed in Beijing. (Note that the OOMs concentrations shown in Fig. 2 of Qiao et al. are as measured by nitrate-CIMS, i.e., without any scaling.)

**9. Line 704. Should $r_{P2-5}$ be $r_{P3-5}$ ?**

**Response:** Yes,  $r_{P2-5}$  has been replaced by  $r_{P3-5}$ .

**Referee 2**: Li and co-workers investigated the new particle formation events in Beijing with a cluster dynamics-multicomponent sectional model. The simulation yields a rich set of information including the time dependent NPF rates, the cluster concentrations, the particle size distributions, and the time- and size-specific particle chemical compositions. They found that the simulations roughly captured the evolution of the observed particle size distributions, and the agreement between the simulation and the observation was improved after the particle growth rates were modulated in the simulation by adjusting the abundance of oxygenated organic molecules. In general, the manuscript is well written and is of broad interest to the readership of Atmospheric Chemistry and Physics. I can recommend publication in Atmospheric Chemistry and Physics after the following comments have been addressed.

**Specific Comments:**

**1.** *Lines 113-114:* "..., but the Gibbs free energy of formation of SA1DMA1 is set to -14.0 kcal/mol at 298 K, ..."

*Please illustrate the potential reasons for setting of the Gibbs free energy of formation of*  $SA_1DMA_1$  *in the present paper.*

**Response**: We thank the reviewer for comment. The formation of  $SA_1DMA_1$  is a critical step in the SA-DMA nucleation mechanism and its Gibbs free energy of formation influences the NPF rates. We set the Gibbs free energy of formation of to -14.0 kJ/mol because with this energy the observed NPF rates in Beijing can be well explained with the SA-DMA mechanism (Cai et al., 2021). Although somewhat arbitrary, this value does lie within the heterodimer formation energy reported by different groups (Myllys et al., 2019; Ortega et al., 2012; Li et al., 2020).

**Revised manuscript, lines 117-118**: "This free energy of SA1DMA1 is the same as that in Cai et al. (2021), which was chosen to improve the agreement between simulated and observed NPF rates."

2. *Lines* 123-124: "More volatile organics may react in the particle phase to form low volatility products and promote particle growth."

The relevant references should be cited for this point of view.

**Response:** The work by Heitto et al. (2022) is cited. This work discussed the effect of oligomerization of organics on particle growth through simulations.

3. *Line 334:* "... (mainly because these processes in the newly formed particles are poorly understood), ... "

The relevant references should be cited for this point of view.

**Response:** We have added three works to the references (Kolesar et al., 2015; Roldin et al., 2014; Yao et al., 2022). In these works, the formation/decomposition of dimers of SOA compounds are discussed. The fitted reaction/decomposition rates differ by orders of magnitude in these studies.

**Revised manuscript, lines 338-340**: "Note that although heterogeneous processes are not considered in the simulation (mainly because these processes in the newly formed particles are poorly understood with highly uncertain rate constants (Kolesar et al., 2015; Roldin et al., 2014; Yao et al., 2022))..."

**Lines 334-336:** "..., OOMs concentration amplification may have similar enhancing effects on particle growth as incorporating heterogeneous reactions which leads to the formation of low volatility products." Are there any restrictions on the amplification of OOMs concentration to enhance the particle growth in the cluster dynamics-multicomponent sectional model?

**Response:** Yes, there should be limitations of OOMs concentration adjustment, which are imposed by the uncertainties of OOMs concentration measurements by nitrate-CIMS. However, at what efficiency nitrate-CIMS detects various compounds still requires further study. Previous works have accounted for the under-detection of OOMs by multiplying the measured OOMs concentration in different VBS bins by different factors, but these factors were obtained empirically by fitting models to the observed particle growth rates (Tröstl et al., 2016). More dedicated studies on OOMs detection efficiency by Riva et al. (2019) indicate it is still difficult to quantify the correction factor for various OOMs species.

Ideally, the OOMs concentration should be amplified based on molecule-specific instrument detection efficiencies. Because this information is still unavailable, in this work we have amplified all OOMs concentration by the same factors to qualitatively explore the possible causes for the simulation-observation discrepancy.

**Figure 6:* Why are the lines in Figure 6(c1) are oscillating while the other lines in Figures 6(c2) and (c3) are smooth?**

**Response:** This is a very good observation. Because it takes a short time for new particles to grow to 2 nm, the composition of 2 nm particles is strongly influenced by the temporal variations of gaseous species, e.g., SA and OOMs. In contrast, it takes a much longer time for particles to grow to 8 nm or 15 nm. As a result, the variations of gaseous species concentrations are smoothed out in the composition of 8 nm and 15 nm. Figs. 6 (c2) and (c3) thus only reflect the overall trend of gas species variation: during the time window examined in Fig 6c, the contribution of SA to particle growth become progressively less than OOMs as time passes.

**Revised manuscript, lines 430-434**: "Compared to the 8 nm and 15 nm particles, the composition of 2 nm particles is oscillatory. Because it takes a short time for new particles to grow to 2 nm, the composition of 2 nm particles reflects the temporal variations of

gaseous species, e.g., SA and OOMs. In contrast, it takes a much longer time for particles to grow to 8 nm or 15 nm. As a result, the variations of gaseous species concentrations are smoothed out in the composition of 8 nm and 15 nm particles."

**Technical corrections:**

*Line 141:* The *i* in some sentences, such as "...  $m_{p,i}$  is the mass of species *i* in the particle, ... ", should be in italics.

Line 222: "Figs. 2a" should be "Fig. 2a". Other similar statements should be corrected.

*Line 261:* "particles formed later in an event has shorter growth time and contribute smaller particles to the mode" should be "particles formed later in an event have shorter growth time and contribute smaller particles to the mode".

*Line 271:* "..., and the average mode diameter normalized by the observed values." should be "..., and the average mode diameter normalized by the observed values, respectively".

**Response to technical corrections:** We thank the reviewer for catching these mistakes. We have corrected the relevant texts.

**References:**

Bianchi, F., Tröstl, J., Junninen, H., Frege, C., Henne, S., Hoyle, C. R., Molteni, U., Herrmann, E., Adamov, A., Bukowiecki, N., Chen, X., Duplissy, J., Gysel, M., Hutterli, M., Kangasluoma, J., Kontkanen, J., Kürten, A., Manninen, H. E., Münch, S., Peräkylä, O., Petäjä, T., Rondo, L., Williamson, C., Weingartner, E., Curtius, J., Worsnop, D. R., Kulmala, M., Dommen, J., and Baltensperger, U.: New particle formation in the free troposphere: A question of chemistry and timing, Science, 352, 1109, 10.1126/science.aad5456, 2016.

Cai, R., Yan, C., Yang, D., Yin, R., Lu, Y., Deng, C., Fu, Y., Ruan, J., Li, X., Kontkanen, J., Zhang, Q., Kangasluoma, J., Ma, Y., Hao, J., Worsnop, D. R., Bianchi, F., Paasonen, P., Kerminen, V. M., Liu, Y., Wang, L., Zheng, J., Kulmala, M., and Jiang, J.: Sulfuric acid-amine nucleation in urban Beijing, Atmos. Chem. Phys., 21, 2457-2468, 10.5194/acp-21-2457-2021, 2021.

Heitto, A., Lehtinen, K., Petäjä, T., Lopez-Hilfiker, F., Thornton, J. A., Kulmala, M., and Yli-Juuti, T.: Effects of oligomerization and decomposition on the nanoparticle growth: a model study, Atmos. Chem. Phys., 22, 155-171, 10.5194/acp-22-155-2022, 2022.

Kolesar, K. R., Chen, C., Johnson, D., and Cappa, C. D.: The influences of mass loading and rapid dilution of secondary organic aerosol on particle volatility, Atmos. Chem. Phys., 15, 9327-9343, 10.5194/acp-15-9327-2015, 2015.

Li, H., Ning, A., Zhong, J., Zhang, H., Liu, L., Zhang, Y., Zhang, X., Zeng, X. C., and He, H.: Influence of atmospheric conditions on sulfuric acid-dimethylamine-ammonia-based new particle formation, Chemosphere, 245, 125554, https://doi.org/10.1016/j.chemosphere.2019.125554, 2020.

Myllys, N., Chee, S., Olenius, T., Lawler, M., and Smith, J.: Molecular-Level Understanding of Synergistic Effects in Sulfuric Acid–Amine–Ammonia Mixed Clusters, J. Phys. Chem. A, 123, 2420-

2425, 10.1021/acs.jpca.9b00909, 2019.

Ortega, I. K., Kupiainen, O., Kurtén, T., Olenius, T., Wilkman, O., McGrath, M. J., Loukonen, V., and Vehkamäki, H.: From quantum chemical formation free energies to evaporation rates, Atmos. Chem. Phys., 12, 225-235, 10.5194/acp-12-225-2012, 2012.

Qiao, X., Yan, C., Li, X., Guo, Y., Yin, R., Deng, C., Li, C., Nie, W., Wang, M., Cai, R., Huang, D., Wang, Z., Yao, L., Worsnop, D. R., Bianchi, F., Liu, Y., Donahue, N. M., Kulmala, M., and Jiang, J.: Contribution of Atmospheric Oxygenated Organic Compounds to Particle Growth in an Urban Environment, Environ. Sci. Technol, 55, 13646-13656, 10.1021/acs.est.1c02095, 2021.

Riva, M., Rantala, P., Krechmer, J. E., Peräkylä, O., Zhang, Y., Heikkinen, L., Garmash, O., Yan, C., Kulmala, M., Worsnop, D., and Ehn, M.: Evaluating the performance of five different chemical ionization techniques for detecting gaseous oxygenated organic species, Atmos. Meas. Tech., 12, 2403-2421, 10.5194/amt-12-2403-2019, 2019.

Roldin, P., Eriksson, A. C., Nordin, E. Z., Hermansson, E., Mogensen, D., Rusanen, A., Boy, M., Swietlicki, E., Svenningsson, B., Zelenyuk, A., and Pagels, J.: Modelling non-equilibrium secondary organic aerosol formation and evaporation with the aerosol dynamics, gas- and particle-phase chemistry kinetic multilayer model ADCHAM, Atmos. Chem. Phys., 14, 7953-7993, 10.5194/acp-14-7953-2014, 2014.

Tröstl, J., Chuang, W. K., Gordon, H., Heinritzi, M., Yan, C., Molteni, U., Ahlm, L., Frege, C., Bianchi, F., Wagner, R., Simon, M., Lehtipalo, K., Williamson, C., Craven, J. S., Duplissy, J., Adamov, A., Almeida, J., Bernhammer, A.-K., Breitenlechner, M., Brilke, S., Dias, A., Ehrhart, S., Flagan, R. C., Franchin, A., Fuchs, C., Guida, R., Gysel, M., Hansel, A., Hoyle, C. R., Jokinen, T., Junninen, H., Kangasluoma, J., Keskinen, H., Kim, J., Krapf, M., Kürten, A., Laaksonen, A., Lawler, M., Leiminger, M., Mathot, S., Möhler, O., Nieminen, T., Onnela, A., Petäjä, T., Piel, F. M., Miettinen, P., Rissanen, M. P., Rondo, L., Sarnela, N., Schobesberger, S., Sengupta, K., Sipilä, M., Smith, J. N., Steiner, G., Tomè, A., Virtanen, A., Wagner, A. C., Weingartner, E., Wimmer, D., Winkler, P. M., Ye, P., Carslaw, K. S., Curtius, J., Dommen, J., Kirkby, J., Kulmala, M., Riipinen, I., Worsnop, D. R., Donahue, N. M., and Baltensperger, U.: The role of low-volatility organic compounds in initial particle growth in the atmosphere, Nature, 533, 527, 10.1038/nature18271, 2016.

Yan, C., Dada, L., Rose, C., Jokinen, T., Nie, W., Schobesberger, S., Junninen, H., Lehtipalo, K., Sarnela, N., Makkonen, U., Garmash, O., Wang, Y., Zha, Q., Paasonen, P., Bianchi, F., Sipilä, M., Ehn, M., Petäjä, T., Kerminen, V. M., Worsnop, D. R., and Kulmala, M.: The role of H2SO4-NH3 anion clusters in ion-induced aerosol nucleation mechanisms in the boreal forest, Atmos. Chem. Phys., 18, 13231-13243, 10.5194/acp-18-13231-2018, 2018.

Yan, C., Yin, R., Lu, Y., Dada, L., Yang, D., Fu, Y., Kontkanen, J., Deng, C., Garmash, O., Ruan, J.,
Baalbaki, R., Schervish, M., Cai, R., Bloss, M., Chan, T., Chen, T., Chen, Q., Chen, X., Chen, Y., Chu,
B., Dällenbach, K., Foreback, B., He, X., Heikkinen, L., Jokinen, T., Junninen, H., Kangasluoma, J.,
Kokkonen, T., Kurppa, M., Lehtipalo, K., Li, H., Li, H., Li, X., Liu, Y., Ma, Q., Paasonen, P., Rantala, P.,
Pileci, R. E., Rusanen, A., Sarnela, N., Simonen, P., Wang, S., Wang, W., Wang, Y., Xue, M., Yang, G.,
Yao, L., Zhou, Y., Kujansuu, J., Petäjä, T., Nie, W., Ma, Y., Ge, M., He, H., Donahue, N. M., Worsnop,
D. R., Veli-Matti, K., Wang, L., Liu, Y., Zheng, J., Kulmala, M., Jiang, J., and Bianchi, F.: The Synergistic
Role of Sulfuric Acid, Bases, and Oxidized Organics Governing New-Particle Formation in Beijing,
Geophys. Res. Lett., 48, e2020GL091944, https://doi.org/10.1029/2020GL091944, 2021.

Yao, L., Garmash, O., Bianchi, F., Zheng, J., Yan, C., Kontkanen, J., Junninen, H., Mazon, S. B., Ehn, M., Paasonen, P., Sipilä, M., Wang, M., Wang, X., Xiao, S., Chen, H., Lu, Y., Zhang, B., Wang, D., Fu,

Q., Geng, F., Li, L., Wang, H., Qiao, L., Yang, X., Chen, J., Kerminen, V.-M., Petäjä, T., Worsnop, D. R., Kulmala, M., and Wang, L.: Atmospheric new particle formation from sulfuric acid and amines in a Chinese megacity, Science, 361, 278, 10.1126/science.aao4839, 2018.

Yao, M., Li, Z., Li, C., Xiao, H., Wang, S., Chan, A. W. H., and Zhao, Y.: Isomer-Resolved Reactivity of Organic Peroxides in Monoterpene-Derived Secondary Organic Aerosol, Environ. Sci. Technol, 56, 4882-4893, 10.1021/acs.est.2c01297, 2022.

Yin, R., Yan, C., Cai, R., Li, X., Shen, J., Lu, Y., Schobesberger, S., Fu, Y., Deng, C., Wang, L., Liu, Y., Zheng, J., Xie, H., Bianchi, F., Worsnop, D. R., Kulmala, M., and Jiang, J.: Acid–Base Clusters during Atmospheric New Particle Formation in Urban Beijing, Environ. Sci. Technol, 55, 10994-11005, 10.1021/acs.est.1c02701, 2021.